# Infinite-World: Scaling Interactive World Models to 1000-Frame Horizons via Pose-Free Hierarchical Memory

Ruiqi Wu [* 1 2 3]  Xuanhua He [* 4 2]  Meng Cheng [* 2]  Tianyu Yang [2]  Yong Zhang [† 2]  Zhuoliang Kang [2]
Xunliang Cai [2]  Xiaoming Wei [2]  Chunle Guo [1 3]  Chongyi Li [1 3]  Ming-Ming Cheng [‡ 1 3]

Project Page: https://rq-wu.github.io/projects/infinite-world/

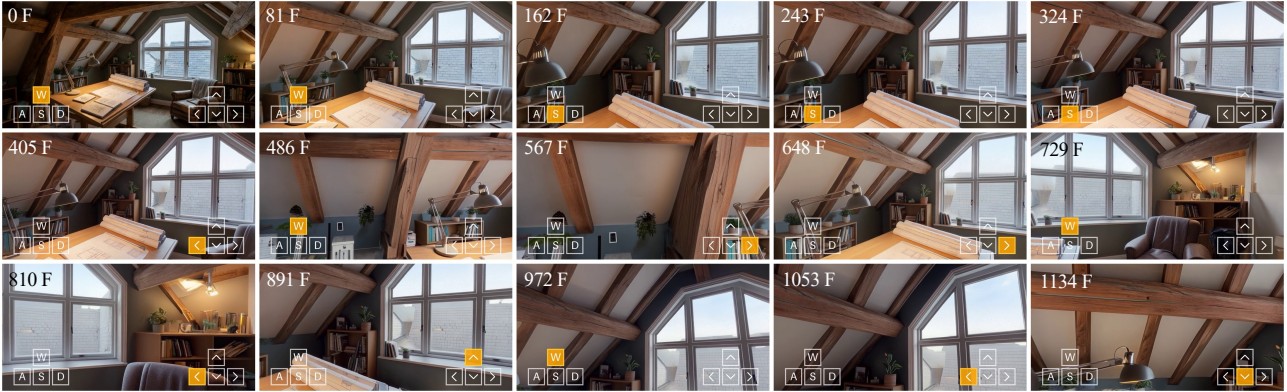

*Figure 1.* **1000-frame interactive world simulation by Infinite-World.** Our model maintains exceptional spatio-temporal consistency and action fidelity over an unprecedented horizon. As the agent explores the indoor environment via keyboard-style controls, *Infinite-World* accurately renders responsive viewpoint changes and preserves global landmarks (e.g., the window and desk arrangement) even after over 1000 frames, demonstrating robust long-range memory and loop-closure capabilities in complex real-world-style scenarios.

## Abstract

We propose **Infinite-World**, a robust interactive world model capable of maintaining coherent visual memory over **1000+ frames** in complex real-world environments. While existing world models can be efficiently optimized on synthetic data with perfect ground-truth, they lack an effective training paradigm for real-world videos due to noisy pose estimations and the scarcity of viewpoint revisits. To bridge this gap, we first introduce a **Hierarchical Pose-free Memory Compressor (HPMC)** that recursively distills historical latents into a fixed-budget representation. By jointly optimizing the compressor with the generative backbone, HPMC enables the model to

autonomously anchor generations in the distant past with bounded computational cost, eliminating the need for explicit geometric priors. Second, we propose an **Uncertainty-aware Action Labeling** module that discretizes continuous motion into a tri-state logic. This strategy maximizes the utilization of raw video data while shielding the deterministic action space from being corrupted by noisy trajectories, ensuring robust action-response learning. Furthermore, guided by insights from a pilot toy study, we employ a **Revisit-Dense Finetuning Strategy** using a compact, 30-minute dataset to efficiently activate the model's long-range loop-closure capabilities. Extensive experiments, including objective metrics and user studies, demonstrate that Infinite-World achieves superior performance in visual quality, action controllability, and spatial consistency.

This work was done during the internship of Ruiqi Wu and Xuanhua He at Meituan. [*]Equal contribution. [†]Project leader. [‡]Corresponding author. [1]NKIARI, Shenzhen Futian [2]Meituan [3]VCIP, CS, Nankai University [4]The Hong Kong University of Science and Technology. Correspondence to: Ming-Ming Cheng <cmm@nankai.edu.cn>.

*Proceedings of the 43ʳᵈ International Conference on Machine Learning*, Seoul, South Korea. PMLR 306, 2026. Copyright 2026 by the author(s).

## 1. Introduction

Recently, World Models (Ha & Schmidhuber, 2018; Yang et al., 2023) have emerged as a focus of research due to their capacity to simulate reality and facilitate interactive

agent-environment control. These models are widely utilized in diverse down-stream domains, such as autonomous driving (Zhang et al., 2025a; Wu et al., 2025b), embodied AI (Wu et al., 2025a; Wang et al., 2025b), and spatial intelligence (Labs, 2024). In particular, the latest breakthroughs in video generation (Wan et al., 2025; Kong et al., 2024; Li et al., 2024) based on diffusion models (Song et al., 2020a;b; Peebles & Xie, 2023) have revealed the potential for modeling the physical world. Recent milestones (Che et al., 2024; He et al., 2025; Li et al., 2025b; HunyuanWorld, 2025) have enabled robust, interactive long-video generation with remarkable visual fidelity. Notably, these models are trained on data from simulation engines like Unreal Engine 5, where perfect ground-truth camera extrinsics and paired blended videos are readily accessible. This provides a clean manifold for the model to learn the action-response mapping.

However, the transition from synthetic domains to real-world scenarios remains a formidable challenge, often referred to as the reality gap of world modeling. While existing frameworks can be efficiently optimized using the perfect labels provided by simulators. It is still challenging to design an effective training paradigm for raw, uncurated video data such as the Sekai dataset (Li et al., 2025d). Current architectures are ill-equipped to handle the following core issues in real-world world modeling:

1. **Inaccurate Pose and Unreliable Control:** Unlike the perfect action labels in simulators, obtaining camera extrinsics from raw video requires estimation, which inevitably introduces errors. This inaccuracy in pose estimation makes precise action-response mapping difficult, leading to degraded controllability.

2. **Scarcity of Viewpoint Revisit Data:** Natural video streams are predominantly "linear", *i.e*, the camera seldom returns to a previously visited location. This lack of dense viewpoint revisits prevents the model from learning the global geometric structure of the environment, making it nearly impossible to acquire long-range spatial memory from standard datasets.

3. **Lack of Efficient Pose-Free Memory Mechanisms:** Standard attention mechanisms suffer from $O(L^2)$ complexity, creating a significant computational bottleneck for long-video generation. Existing solutions either depend heavily on explicit, often inaccurate, camera poses for frame retrieval (Xiao et al., 2025; Yu et al., 2025) or employ simple downsampling (Zhang et al., 2025b) that incurs severe information loss. A memory mechanism that is both pose-free and computationally efficient remains elusive.

In this paper, we present **Infinite-World**, a robust interactive world model that breaks these barriers to achieve stable simulation over an unprecedented horizon of **1000+ frames**. Specifically, we design a novel memory mechanism to support training and inference with extremely long contexts. An effective training paradigm is also proposed to achieve robust action controllability and history memory on real-world data.

First, we tackle the memory and computational bottleneck through a **Hierarchical Pose-free Memory Compressor (HPMC)**. As illustrated in Figure 2(a), the HPMC employs a two-stage recursive distillation pipeline to transform raw historical latents into a fixed memory budget $T_{max}$. Historical latents first undergo **Local Compression** to capture fine-grained, short-range dynamics. For long-horizon sequences exceeding the processing threshold, the model initiates a **Global Compression** stage by utilizing an adaptive sliding-window sampling mechanism. These sampled chunks are recursively distilled and concatenated into a unified global representation, ensuring the final memory remains strictly within the $T_{max}$ limit. This hierarchical structure maintains a constant computational footprint and prevents memory drift during thousand-frame simulations. During training, the compressor is jointly optimized with the DiT backbone, learning to autonomously distill salient cues for future synthesis. Consequently, our model eliminates the need for external pose metadata, achieving long-range spatial consistency in a purely data-driven, pose-free manner.

Second, we propose an **Uncertainty-aware Action Labeling** strategy to handle the noise in real-world trajectories. As shown in Figure 2(b), this mechanism discretizes continuous motion into a tri-state logic: *No-operation*, *Discrete Action*, and *Uncertain*. By explicitly labeling ambiguous, low-SNR motion as "Uncertain" rather than discarding these samples, we maximize the utilization of raw video data while preserving the temporal continuity essential for training video models. This strategy effectively shields the deterministic action space from being corrupted by jitter or slow-drifting poses, ensuring robust and responsive action-response learning despite imperfect pose estimations.

Furthermore, we propose a **Revisit-Dense Finetuning Strategy** based on the fundamental requirements for spatial memory identified in our pilot study. Our experiments reveal that loop-closure capability can be "activated" with surprisingly low data volume, whereas exceeding the temporal window encountered during training leads to catastrophic memory collapse. We conclude that the bottleneck for long-horizon modeling lies in the duration and topological density of individual trajectories rather than total data quantity. Consequently, we curate a 30-minute Revisit-Dense Dataset (RDD) to efficiently activate the model's long-range consistency and loop-closure capabilities.

The main contributions of this work are summarized as follows:

- We propose a Hierarchical Pose-Free Memory Compressor that scales interactive world modeling to 1000+ frames with bounded computational cost and without requiring explicit camera poses.

- We design an effective training framework for real-world raw videos, featuring an uncertainty-aware action labeling to mitigate the impact of noisy pose estimations and a Revisit-Dense Finetuning Strategy to activate long-range memory capabilities.

- Extensive experiments, including objective metrics, a user study, and visual comparisons, demonstrate that Infinite-World achieves superior performance in visual quality, action controllability, and memory consistency.

## 2. Related Work

### 2.1. Interactive World Models

Recent advancements in video generation (Zheng et al., 2024; Wan et al., 2025; Kong et al., 2024) have demonstrated a remarkable capacity to synthesize sequences that adhere to physical laws (Kang et al., 2024) and exhibit strong 3D consistency (Li et al., 2024), revealing their potential to serve as foundational world models (Ha & Schmidhuber, 2018; Yang et al., 2023). Building upon these generative priors, several studies have extended such models to achieve action-responsiveness and long-horizon video generation. Gamegen-X (Che et al., 2024) pioneered train an interactive video model on game data. Matrix-Game 2.0 (He et al., 2025) build a large-scale data pipeline using UE5 and GTA5 to obtain synchronized video-action pairs, enabling a world model that responds to keyboard and mouse inputs. Similarly, Hunyuan-GameCraft (Li et al., 2025b) utilized high-quality AAA game footage to scale up interactive capabilities. To extend temporal coherence, RELIC (Hong et al., 2025) distilled long-video generative priors and utilized synthetic data to achieve stable 20-second simulations.

Despite these successes, a significant domain gap remains between synthetic environments and the complexity of the real world. To address this, the Sekai dataset (Li et al., 2025d) introduced a vast collection of first-person world exploration videos from the web. Subsequent works, such as HY-World- 1.5 (HunyuanWorld, 2025), combined Sekai data with synthetic sequences to enhance generalizability, while Yume 1.0 (Mao et al., 2025b), Yume 1.5 (Mao et al., 2025a), and MagicWorld (Li et al., 2025a) focused on training directly on real-world distributions. However, compared to synthetic and game data, real-world data introduces complex dynamic scenes and erratic camera jitter. The difficulty in obtaining accurate camera poses severely limits action-controllability, and the scarcity of viewpoint revisits makes it challenging for models to acquire long-context

memory. While Genie-3 (Ball et al., 2025) has demonstrated state-of-the-art performance in real-world simulation, its closed-source nature leaves the technical details of its training paradigms and architectures inaccessible to the research community. Our work, Infinite-World, aims to fill this gap by providing an open and efficient framework for long-horizon real-world interactive modeling.

### 2.2. Context-Memory in Long Video Generation

Quadratic attention complexity often leads to computational collapse in long-video generation. To mitigate this, geometry-dependent methods use camera poses to filter context, such as 3D scene reconstruction (Li et al., 2025c; Huang et al., 2025) or Field-of-View (FOV) overlap retrieval (Xiao et al., 2025; Yu et al., 2025), a strategy also adopted by HY-World-1.5 (HunyuanWorld, 2025). However, these methods are sensitive to pose estimation errors, which are prevalent in real-world data.

Alternatively, compression-based methods like Hunyuan-GameCraft (Li et al., 2025b), RELIC (Hong et al., 2025), and Yume 1.5 (Mao et al., 2025a) employ uniform temporal downsampling inspired by FramePack (Zhang et al., 2025b), which slows latent growth but incurs severe information loss. SlowFastGen (Hong et al., 2024) proposes an implicit strategy via inference-time LoRA finetuning to memorize scenes, but at the cost of significant computational overhead. Concurrently, PFP (Zhang et al., 2025c) utilizes a two-stage strategy by pretraining a memory compressor with a frame preservation objective to optimize retrieval fidelity. However, PFP compresses the context through a high but fixed compression ratio, its memory footprint still scales linearly with sequence length. In contrast, **Infinite-World** introduces a Hierarchical Pose-free Memory Compressor that recursively compresses latents into a fixed-budget representation. We achieve over 1000-frame consistency without explicit geometric priors or inference-time optimization.

## 3. Methodology

In this section, we formally present the architecture and learning paradigm of **Infinite-World**. Our goal is to enable an interactive world model to maintain a coherent state over a 1000-frame horizon while learning from noisy, real-world data. The framework consists of three core components: (1) a hierarchical pose-free memory compressor for constant-complexity context modeling; (2) an uncertainty-aware action labeling module for robust motion control; and (3) a revisit-dense training objective.

### 3.1. Hierarchical Pose-free Memory Compressor

To achieve a stable 1000-frame simulation with a constant computational cost, we propose the **Hierarchical Pose-free**

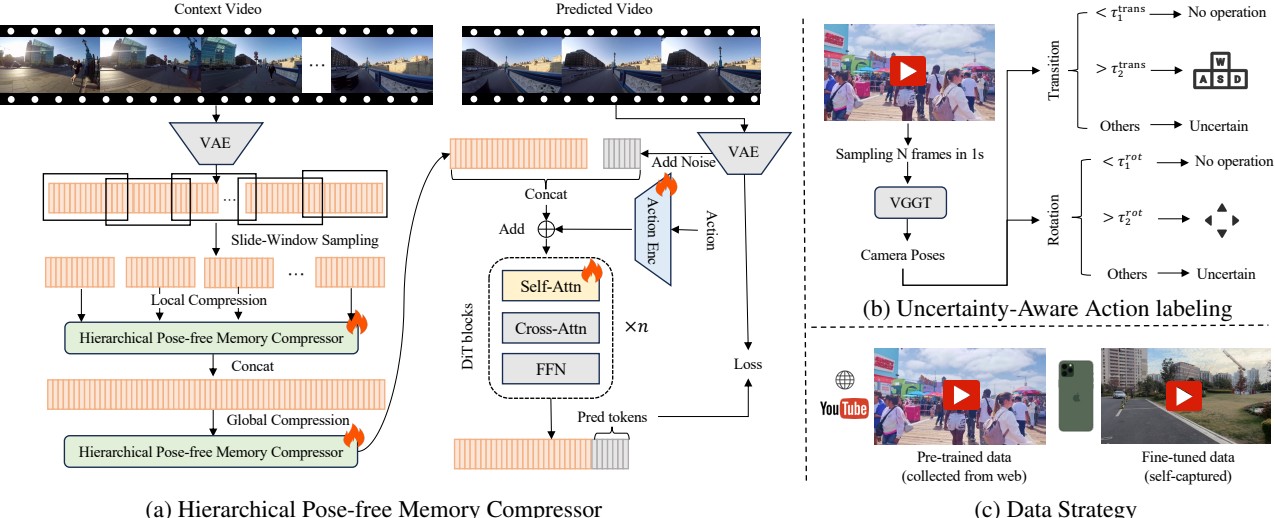

(a) Hierarchical Pose-free Memory Compressor

(b) Uncertainty-Aware Action labeling

(c) Data Strategy

*Figure 2.* **Overview of Infinite-World architecture. (a) Hierarchical Pose-free Memory Compressor:** The Hierarchical Pose-free Memory Compressor (HPMC) recursively compresses raw historical latents into a fixed memory budget via hierarchical compression with local and global stage. The compressor is jointly optimized with the DiT backbone to autonomously anchor generations in the distant past with constant computational cost. **(b) Uncertainty-Aware Action labeling:** Continuous poses are decoupled into translation and rotation primitives. A tri-state logic filters out "Uncertain" motion to ensure robust action-response learning. **(c) Data Strategy:** Pre-training on open-domain video is followed by finetuning on a revisit-dense dataset to activate 1000-frame memory consistency.

**Memory Compressor (HPMC)** as illustrated in Figure 2(a). HPMC transforms historical latents into a fixed memory budget through two operational modes based on the context length: **Direct Compression** for short-range sequences and **Hierarchical Compression** for long-range exploration.

**Mode 1: Short-Horizon Direct Compression.** For sequences where the context length is within a manageable threshold ($L \le k \cdot T_{max}$), we apply a temporal encoder $f_\phi$ directly to the raw latents $\mathbf{z}_{1:L}$. This stage acts as a high-fidelity filter that reduces the temporal resolution by a factor of $k = 4$, producing compressed tokens $\mathbf{z}_{com}$:

$$\mathbf{z}_{com} = f_\phi(\mathbf{z}_{1:L}), \quad \mathbf{z}_{com} \in \mathbb{R}^{\frac{L}{4} \times d} \quad (1)$$

where $d$ is the model's hidden dimension. This representation is used directly as the conditioning context for the DiT backbone.

**Mode 2: Long-Horizon Hierarchical Compression.** When the exploration horizon $L$ exceeds the memory budget, we introduce a hierarchical compression mechanism to prevent memory drift. As shown in Figure 2(a), raw latents are first partitioned into $N$ overlapping chunks via a sliding-window $\mathcal{W}$ with size $W$ and dynamic stride $S$. Each chunk undergoes a first-stage **local compression** to extract salient spatio-temporal cues. These intermediate tokens are then concatenated and subjected to a second-stage **global compression** using the same encoder $f_\phi$:

$$\mathbf{z}_{com} = f_\phi \left( \text{Concat} \left( \{ f_\phi(\text{Chunk}_i) \}_{i=1}^N \right) \right) \quad (2)$$

This recursive structure compresses wide-span history into a unified global representation $\mathbf{z}_{com}$ while maintaining a strictly bounded memory footprint.

**Context Injection to Diffusion Transformer.** To guide the generation process, the DiT input is a temporal concatenation of the compressed history $\mathbf{z}_{com}$, the last-frame latent serving as local memory $\mathbf{z}_{loc}$, and the noisy target latents $\mathbf{z}_t$. A binary mask $\mathbf{m}$ is appended to the sequence to distinguish the context from the denoising target.

**Joint Optimization and Pose-free Anchoring.** A key distinction of our HPMC is that the compressor $f_\phi$ is **jointly optimized** end-to-end with the DiT backbone (indicated by the fire icons in Figure 2(a)). By training the summarizer to minimize the generation loss of future frames, the model learns to autonomously identify and preserve the most relevant historical cues for loop-closure. Consequently, the HPMC eliminates the need for external pose metadata or explicit geometric priors, achieving long-range spatial consistency in a purely data-driven, **pose-free** manner.

### 3.2. Uncertainty-Aware Action Labeling and Encoding

To bridge the gap between continuous control and noisy real-world trajectories, we propose an uncertainty-aware labeling and encoding mechanism that transforms raw motion into a discrete action space $\mathcal{A} = \{a_{trans}, a_{rot}\}$.

**Motion Decoupling and Tri-state Labeling.** Given a real-world video sequence, we first estimate the relative camera pose changes $\Delta P$ between consecutive frames using an off-the-shelf pose estimator (Wang et al., 2025a). We decouple the 6-DoF pose change into translation magnitude

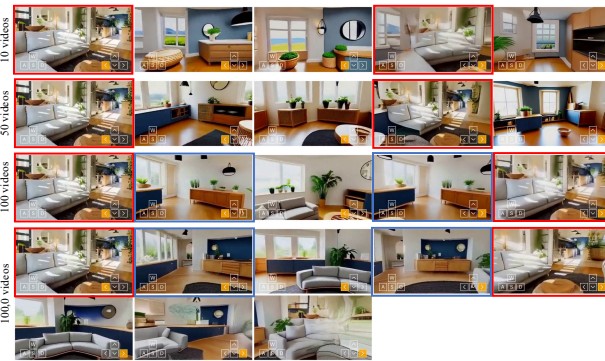

*Figure 3.* **Pilot study on spatial memory.** Each image is the first frame of a chunk. (Rows 1-3): Memory activation saturates at only 100 training sequences, with loop-closure capability effectively established. (Row 4): Catastrophic collapse occurs during 1000-frame inference as the horizon significantly exceeds the 4-chunk training context.

$\|\Delta P_{trans}\|$ and rotation magnitude $\|\Delta P_{rot}\|$. To prevent catastrophic misalignment during training, we introduce a tri-state logic to categorize motion intensity using two domain-specific thresholds, $\tau_1$ (noise floor) and $\tau_2$ (action trigger). For each dimension, the label $a$ is assigned as follows:

$$a = \mathcal{M}(\Delta P; \tau_1, \tau_2) = \begin{cases} \text{No-operation} & \text{if } \|\Delta P\| < \tau_1 \\ \text{Discrete Action} & \text{if } \|\Delta P\| > \tau_2 \\ \text{Uncertain} & \text{otherwise} \end{cases} \quad (3)$$

The *Discrete Action* state is mapped to semantic directions: {W, A, S, D} for translation and {Left, Right, Up, Down} for rotation. We explicitly retain the *Uncertain* state to represent motion with low signal-to-noise ratios, preventing slow-moving data from being incorrectly categorized as "No-operation" while shielding action labels from noisy jitter.

**Temporally-Aligned Action Injection.** Simultaneously, the **Action Encoder** transforms raw movement and viewpoint sequences into an embedding $\mathbf{e}_{act}$. To achieve temporal alignment, the encoder employs two 1D-convolutional layers with a stride of 2, resulting in a $4\times$ downsampling rate that strictly matches the latent resolution of the compressed visual history ($k = 4$). This ensures the active segment of $\mathbf{e}_{act}$ is identical in length to the patched target latents $\mathbf{z}_t$. After zero-padding for historical segments, the final embedding is element-wise added to the video tokens:

$$\mathbf{x} = \text{Patch}(\mathbf{z}_{com}, \mathbf{z}_{loc}, \mathbf{z}_t, \mathbf{m}) + \mathbf{e}_{act} \quad (4)$$

This design allows action signals to directly modulate the noisy latent space with precise temporal synchronization and minimal overhead.

### 3.3. Revisit-Dense Finetuning Strategy

The design of our data strategy is motivated by a pilot study aimed at identifying the minimum requirements for spatial memory. As illustrated in Figure 3, we train a simplified DiT-based generator using synthetic 3D scenes, where historical latents are provided as concatenated context. Our empirical findings reveal two critical insights:

**1. High Sample Efficiency of Memory:** The ability to achieve loop-closure can be activated with low data amount. While 10 to 50 video sequences already enable the model to reference historical cues (red pairs in row 1–2 of Figure 3), increasing the diversity to 100 sequences is sufficient to establish robust spatial memory with accurate 3D consistency (red and blue pairs in row 3 of Figure 3). Scaling the data to 1,000 sequences yields marginal gains, suggesting that memory acquisition is more dependent on topological variety than absolute quantity.

**2. Context-Bound Extrapolation:** We observe a strict coupling between memory stability and the training temporal window. When a model trained on a maximum context of 4 chunks is tasked with inference over longer sequences (e.g., 6 chunks), its memory mechanism undergoes a catastrophic collapse, leading to significant visual drift and hallucinations.

Based on these observations, we conclude that the bottleneck for long-horizon world modeling is the *duration* and *topological density* of trajectories rather than data *quantity*. We thus employ a two-stage strategy:

**1. Open-Domain Pre-training.** We first pre-train the model on a large-scale real-world dataset to learn diverse visual priors and local dynamics. We utilize relatively short video sequences, as internet-collected data typically lacks scenarios involving long-term viewpoint revisits or complex loop-closure.

**2. Memory Activation via RDD.** We utilize a compact Revisit-Dense Dataset (RDD) to activate the model's spatial memory. By leveraging the high sample efficiency observed in our pilot study, we curate a small set of revisit-dense videos with long time-duration. This allows us to bridge the reality gap and achieve stable 1000-frame loop-closure at a practical cost.

## 4. Experiments

### 4.1. Experimental Setup

#### 4.1.1. IMPLEMENTATION DETAILS

Our model is built on Wan-2.1-1.3B (Wan et al., 2025). We adopt the training strategy of (Li et al., 2025b) that uses diverse lengths of context frames for mix training. During the pre-training stage, we exclusively employ the direct

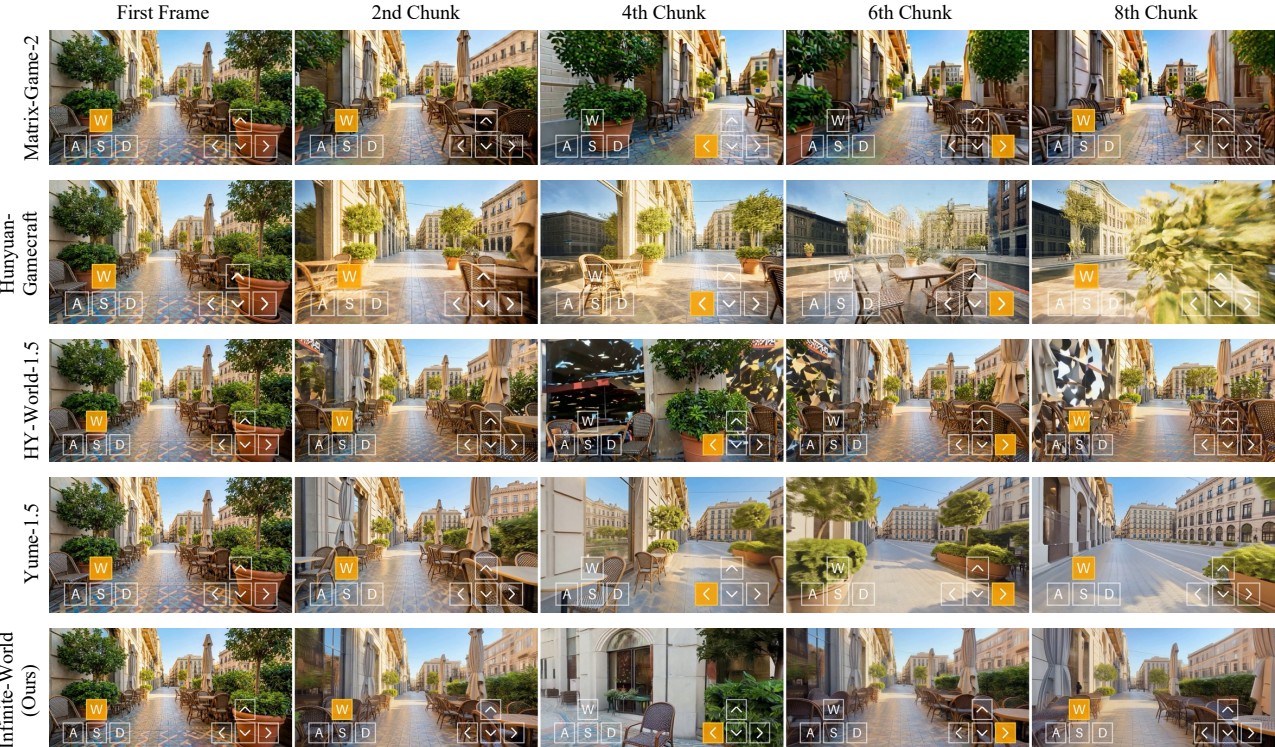

*Figure 4.* Visual comparison between the proposed Infinite-World and four baselines. Notice the visual consistency between 2nd chunk and 6th chunk, and 8th chunk is the zoom-in view of first frame.

compression, with the history videos capped at a maximum length of four temporal chunks. For fine-tuning, we expand the temporal receptive field by sampling historical frames with a duration spanning from one image to 16 chunks. The hierarchical compression is used for a higher efficiency. Both stages are trained at a resolution of 480P. We use the AdamW optimizer with a constant learning rate of $1 \times 10^{-5}$. The Hierarchical Memory Compressor employs a 3D-ResNet structure to compress latents, achieving a temporal compression factor of $k = 4$. In our implementation, we utilize $N = 5$ overlapping chunks with a constant window size of $W = 64$ frames. The sliding stride $S$ is dynamically adjusted based on the total latent length to ensure the entire historical horizon is fully covered. This hierarchical design supports a context of up to 320 frames, which is recursively compressed into a fixed memory budget of $T_{max} = 20$. All experiments are conducted on 16×NVIDIA H800 GPUs.

### 4.1.2. DATASET

**Pre-training Set:** A large-scale video dataset collected from the internet, consisting of over 30 hours of first-person and exploration footage. This dataset provides a diverse prior for natural dynamics but lacks dense viewpoint revisits.

**Finetuning Set (RDD):** It contains 30 minutes of high-quality, long-duration videos characterized by frequent loop-closures. To ensure high visual fidelity and stability, we recorded the footage using an iPhone 17 Pro in Action

Mode, effectively minimizing camera jitter and motion blur. Despite its small scale, RDD is sufficient to activate the model's 1000-frame consistency.

### 4.2. Benchmarks

#### 4.2.1. DATA CONSTRUCTION

To evaluate open-domain interactive performance, we establish a benchmark comprising 100 diverse scenarios. Specifically, we utilize Gemini (Gemini Team, 2023) to generate 100 text prompts covering *Indoor, Street, Nature,* and *Fantasy* domains. For each prompt, we employ the Nanobanana (Google, 2025) text-to-image model to synthesize a high-fidelity initial frame as the world state. To assess action-responsiveness, we manually design 10 representative long action trajectories (16 chunks). Every generated scene is assigned one of these handcrafted trajectories to simulate long-horizon exploration. This setup ensures a rigorous evaluation of the model's ability to maintain spatial consistency and action fidelity across varied environments.

#### 4.2.2. BASELINES

We evaluate **Infinite-World** against several state-of-the-art interactive world models: **HY-World 1.5** (HunyuanWorld, 2025), which utilizes FOV-based attention mechanism to achieve memorization, and trained on mixed synthetic and real-world data; **Hunyuan-GameCraft** (Li et al., 2025b),

*Table 1.* **Quantitative comparison on our long-horizon interactive benchmark.** We report objective quality metrics from VBench and multi-dimensional subjective scores from our user study. **Bold** and underline indicate the first and second best results, respectively.

| Model | VBench | | | | | User Study | | | |
|---|---|---|---|---|---|---|---|---|---|
| | Mot. Smo.↑ | Dyn. Deg.↑ | Aes. Qual.↑ | Img. Qual.↑ | Avg. Score↑ | Memory ↓ | Fidelity ↓ | Action ↓ | ELO Rating ↑ |
| Hunyuan-GameCraft | 0.9855 | 0.9896 | 0.5380 | 0.6010 | 0.7785 | 2.67 | 2.49 | 2.56 | 1311 |
| Matrix-Game 2.0 | 0.9788 | **1.0000** | 0.5267 | **0.7215** | 0.8068 | 2.98 | 2.91 | 1.78 | 1432 |
| Yume 1.5 | 0.9861 | 0.9896 | **0.5840** | 0.6969 | **0.8141** | 2.43 | 1.91 | 2.47 | 1495 |
| HY-World-1.5 | **0.9905** | **1.0000** | 0.5280 | 0.6611 | 0.7949 | 2.59 | 2.78 | **1.50** | 1542 |
| **Infinite-World** | 0.9876 | **1.0000** | 0.5440 | 0.7159 | 0.8119 | **1.92** | **1.67** | 1.54 | **1719** |

optimized on AAA game footage for high-fidelity rendering; **Yume 1.5** (Mao et al., 2025a), a real-world model employing uniform temporal downsampling for memorization; and **Matrix-Game 2.0** (He et al., 2025), a neural simulator trained on extensive UE5/GTA5 synthetic pipelines to prioritize 3D consistency.

#### 4.2.3. EVALUATION METRICS

To evaluate our method and the comparison baselines across multiple dimensions, we adopt a hybrid evaluation framework that combines automated generative metrics with human-centric preference rankings.

**Automated Quality Assessment.** We utilize the VBench suite (Huang et al., 2024; Zheng et al., 2025) to quantify the overall quality of generated trajectories. Our evaluation focuses on *motion smoothness*, *dynamic degree*, and *imaging quality* to capture the physical realism and visual fidelity of the simulation. Notably, we deliberately exclude *subject consistency* and *background consistency* from our assessment. As an interactive world model designed for scene exploration, our sequences involve significant viewpoint transitions where both the subject and background are inherently non-stationary.

**User Study.** Given the limitations of automated metrics in assessing interactive experiences, we conduct a large-scale user study to obtain fine-grained evaluations across multiple performance dimensions, while maintaining an ELO rating system to reflect overall preference. In each trial, participants are presented with randomized pairs of results from different methods and are asked to rank them based on three key dimensions: (1) **Memory Consistency**, focusing on long-term world stability and loop-closure capability; (2) **Visual Fidelity**, assessing realism and the mitigation of long-horizon cumulative errors; (3) **Action Responsiveness**, measuring the alignment between control signals and visual outputs. Finally, participants select the overall superior simulation to update the ELO scores.

#### 4.3. Comparisons

#### 4.3.1. QUANTITATIVE RESULTS

**Objective Metrics.** As reported in Table 1, Infinite-World achieves the best or second-best performance across all

VBench dimensions. While Yume 1.5 holds a marginal lead in average score (0.8141 vs. 0.8119), this is primarily driven by its superior *aesthetic quality* (0.5840), which is mostly a consequence of its significantly larger parameter scale (5B vs. our 1.3B). Furthermore, we observe that Yume 1.5's high benchmark scores are partly due to its limited action control. As reflected in the subsequent visual comparison, Yume 1.5 frequently defaults to simple "move forward" trajectories, failing to execute complex viewpoint transitions. By staying within a narrow field-of-view relative to the initial frame, it avoids the challenge of generating novel out-of-view content.

**User Study.** While objective metrics offer a preliminary assessment, our human subjective evaluation provides a more comprehensive reflection of the model's interactive performance. Infinite-World demonstrates decisive superiority, achieving a dominant ELO rating of 1719. This represents a substantial 177-point lead over the next best-performing model, HY-World-1.5 (1542).

The fine-grained rankings pinpoint our technical advantages. Infinite-World secures the top rank in *Memory Consistency* (1.92) and *Visual Fidelity* (1.67), significantly outperforming Matrix-Game 2.0 and HY-World-1.5. This proves that our Hierarchical Pose-Free Memory (HPMC) effectively mitigates error accumulation over 1000-frame horizons. Crucially, our model achieves a state-of-the-art *Action Responsiveness* rank of 1.54, which is comparable to the performance of HY-World-1.5 (1.50). It is noteworthy that while HY-World-1.5 relies on perfectly-labeled synthetic data for training, **Infinite-World** achieves the same level of responsiveness using noisy, raw real-world videos. This success validates that our uncertainty-aware action labeling effectively bridges the reality gap, delivering immediate and accurate feedback even when trained on imperfect real-world trajectories.

#### 4.3.2. QUALITATIVE RESULTS

We present a visual comparison against state-of-the-art models in Figure 4. Matrix-Game 2.0 achieves high visual fidelity through chunk-wise I2V generation but lacks a mechanism for out-of-view memory. Hunyuan-GameCraft maintains coarse scene persistence, but fails to preserve fine-grained structural details over long horizons. While HY-

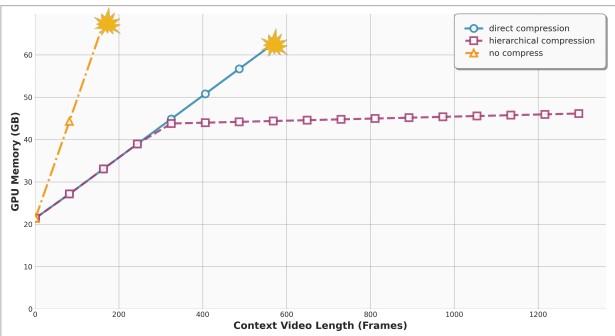

*Figure 5.* Comparison of memory consumption relative to context video length on an 80GB H800. Our hierarchical compression achieves near-constant memory overhead after an initial growth phase, whereas non-compressed baselines suffer from rapid memory exhaustion.

World-1.5 excels in short-term consistency, it suffers from significant error accumulation, leading to ghosting artifacts and structural distortions. Yume 1.5, trained on real-world data, is hampered by a motion distribution bias. The dominance of forward-moving trajectories in raw video causes a collapsed "forward-moving" tendency regardless of the input command, making it difficult to execute viewpoint revisits or verify out-of-view memory. In contrast, Infinite-World overcomes these biases via uncertainty-aware action labeling, achieving responsive action control. Our model preserves global landmarks after hundreds of frames, successfully performing long-range loop closures where baselines fail, validating the effectiveness of our hierarchical memory compressor in activating spatial reasoning.

## 4.4. Ablative Study

In this section, we analyze the computational effectiveness of Hierarchical Pose-free Memory Compressor (HPMC), and evaluate the contribution of Revisit-Dense Dataset (RDD) finetuning and Uncertainty-aware Action Labeling (UAL). Since variants without RDD finetuning are primarily trained on short videos, we conduct all comparative evaluations at a uniform temporal horizon of five chunks to ensure a fair and consistent assessment across all configurations.

### 4.4.1. EFFICIENCY ANALYSIS OF HISTORY COMPRESSION

To evaluate the impact of our HPMC on long-horizon inference, we compare GPU memory consumption across three configurations: no compression, direct compression, and our hierarchical compression. As illustrated in Figure 5, without compression, memory usage scales linearly and triggers an out-of-memory error at over 180 frames. While direct compression reduces the growth rate, it still exhibits a linear trend. In contrast, our hierarchical compression demonstrates a distinct memory plateau, stabilizing at approximately 45GB. This sub-linear scaling ensures that computational overhead remains bounded even as the explo-

*Table 2.* **Ablation results.** *UAL* denotes Uncertain-Aware Labeling; *RDD FT* denotes Revisit-Dense Dataset finetuning.

| Configurations | Fidelity ↓ | Memory ↓ | Action ↓ |
|---|---|---|---|
| Baseline | 2.10 | 2.40 | 2.95 |
| + UAL | 1.78 | 2.17 | 2.17 |
| + RDD FT | 1.83 | 1.83 | 1.61 |
| **Full Model** | **1.75** | **1.69** | **1.38** |

ration horizon extends to 1300 frames and beyond.

### 4.4.2. EFFECTIVENESS OF RDD FINETUNING

As shown in Table 2, RDD finetuning is the primary driver for "activating" long-range spatial memory. Comparing the Baseline to the +RDD FT variant, the *Memory Consistency* rank improves significantly from 2.40 to 1.83. Beyond memory, RDD finetuning also notably enhances action control (improving from 2.95 to 1.61). This dual benefit stems from the nature of the RDD dataset, which maintains stable and consistent action magnitudes. By training on these high-topological-density trajectories, the model learns a more robust and smooth action-response mapping compared to noisy, linear open-domain videos.

### 4.4.3. EFFECTIVENESS OF UAL

The inclusion of our Uncertainty-aware Action Labeling (UAL) yields consistent improvements in *Action Responsiveness* across different training stages. As reported in Table 2, the action rank improves from 2.95 to 2.17 when UAL is added to the baseline. A similar trend is observed in the full model. These results demonstrate that our tri-state logic effectively shields the action space from pose estimation noise, regardless of the underlying dataset, ensuring immediate and accurate feedback during interaction.

## 5. Conclusion and Future Works

In this paper, we presented **Infinite-World**, an interactive world model achieving coherent simulation over a 1000-frame horizon. By integrating a **Hierarchical Pose-free Memory Compressor** with **Uncertainty-aware Action Labeling**, we successfully addressed the challenges of computational complexity and noisy real-world pose estimations. Furthermore, our RDD strategy demonstrates that long-range loop-closure can be activated with minimal data.

While this work focuses on establishing long-term memory in real-world scenarios, several areas remain for future exploration. To further mitigate cumulative drift and visual degradation, techniques such as self-forcing or refined noise schedules could be employed. Additionally, model distillation and scaling to larger backbones offer pathways to improve inference speed and visual fidelity. We hope Infinite-World serves as a robust foundation for future real-world neural simulators.

## Impact Statement

This paper presents work whose goal is to advance the field of Machine Learning. Our research into long-horizon interactive world modeling has potential implications for the reliability of autonomous systems and the efficiency of large-scale generative model training. While we highlight the benefits for simulation and embodied AI, we also acknowledge the broader ethical responsibility to prevent the misuse of high-fidelity video generation technologies.

## Acknowledgment

This work is supported in part by the NSFC (NO. 62225604) Shenzhen Science and Technology Program (JCYJ20240813114237048), Tianjin Natural Science Foundation Project (25ZXRGGX00290, 24JCJQJC00020, 25JC-QNJC01390), the National Natural Science Foundation of China (62306153, 62225604), the Young Elite Scientists Sponsorship Program by CAST (YESS20240686), and the Fundamental Research Funds for the Central Universities (Nankai University, 63253223, 63253219).

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

## A. Detailed User Study Protocol

To evaluate the interactive performance and long-term consistency of **Infinite-World** compared to state-of-the-art baselines, we conducted a comprehensive human subjective evaluation.

### A.1. Evaluation Interface and Procedure

We developed a side-by-side comparison web interface as shown in Figure 6. For each evaluation session, the system randomly selects two methods from a pool of five candidates (our **Infinite-World** and four baseline models: Hunyuan-GameCraft, Matrix-Game 2.0, Yume 1.5, and HY-World-1.5).

Participants are presented with two synchronized videos generated from identical initial frames and interactive action sequences. To ensure a double-blind study, the methods are anonymized as "Method A" and "Method B," with their horizontal positions randomized for each trial.

### A.2. Participant Selection and Quality Control

We recruited 30 volunteers with backgrounds in computer vision or interactive media to ensure the reliability of the subjective feedback. Since our benchmark involves long-range interactive sequences, each comparison requires a significant amount of time for thorough observation (approximately 3 minutes per trial).

To prevent evaluator fatigue and ensure high assessment quality, we limited each participant to 10 comparison trials. This design choice ensures that every judgment is made with high focus, resulting in a total of 300 high-quality pairwise comparisons.

### A.3. Evaluation Dimensions

Participants rate the two methods across three fine-grained dimensions using a 5-point scale:

- **Visual Quality:** Evaluates temporal stability and aesthetic fidelity.

- **Memory Ability:** Specifically assesses scene consistency during loop-closure (returning to previously visited view-points).

- **Action Response Ability:** Measures the accuracy and immediacy of world state evolution in response to action inputs.

### A.4. Statistical Analysis

To synthesize these sparse pairwise comparisons into a global ranking, we employ the **ELO rating system**. The ELO score provides a robust measure of overall human preference by accounting for the relative strength of the randomly paired models. Additionally, we report the **mean numerical rank** for each of the three fine-grained dimensions, where a lower value indicates superior performance as judged by the volunteers.

## B. More Visual Comparisons

We illustrate additional visual comparisons in Figure 7 and Figure 8. To provide a clearer and more focused assessment of long-horizon capabilities, we restrict this visual analysis to a direct comparison between our **Infinite-World** and the second-best performing baseline from our user study, **HY-World-1.5**.

As evident in these figures, our method demonstrates superior long-term stability. It effectively maintains **memory consistency** across extended temporal horizons, successfully preserving key scene structures during loop-closures. In contrast, HY-World-1.5 tends to accumulate significant visual artifacts and suffers from severe structural degradation as the simulation progresses.

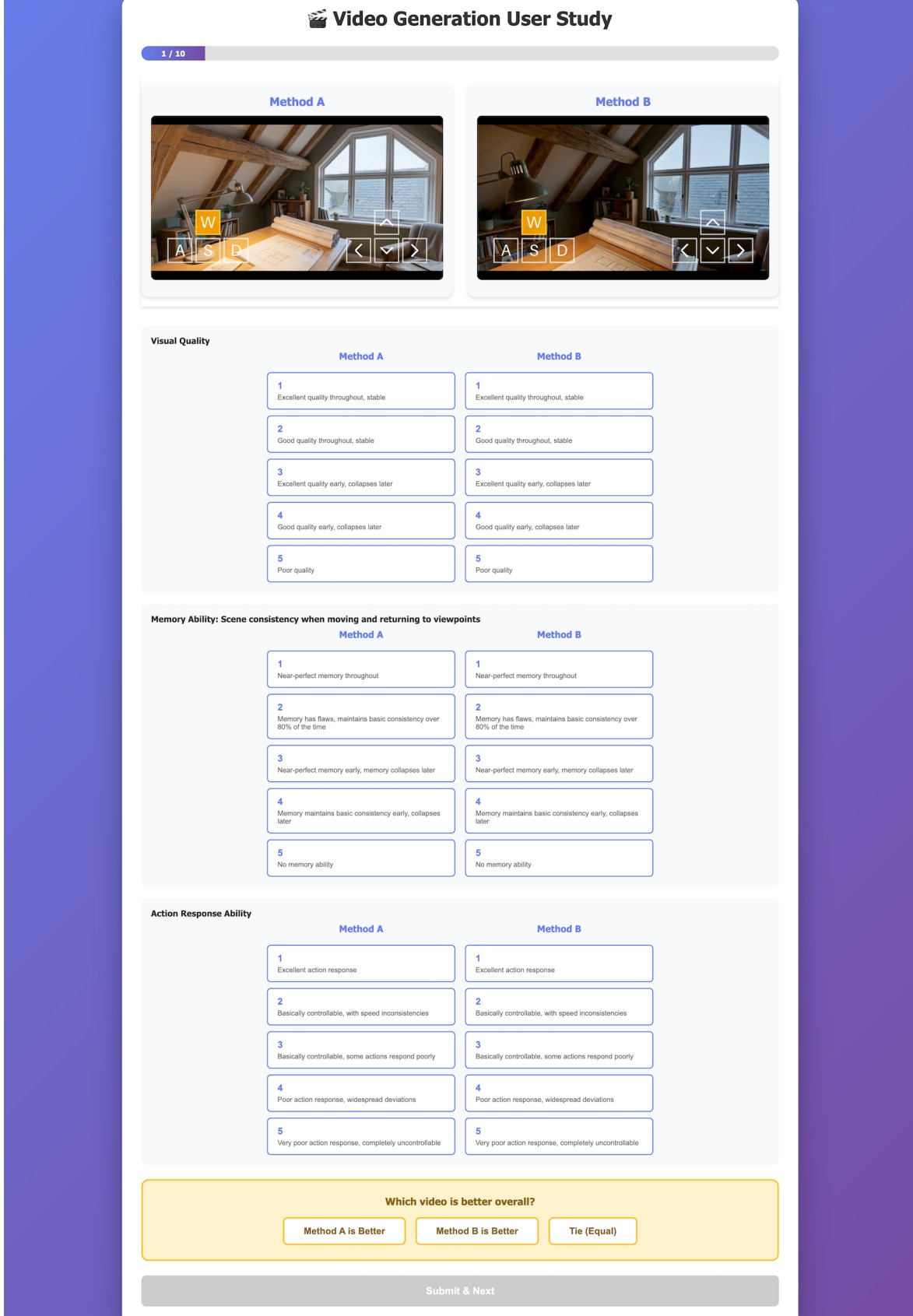

*Figure 6.* Screenshots of our user study webpage.

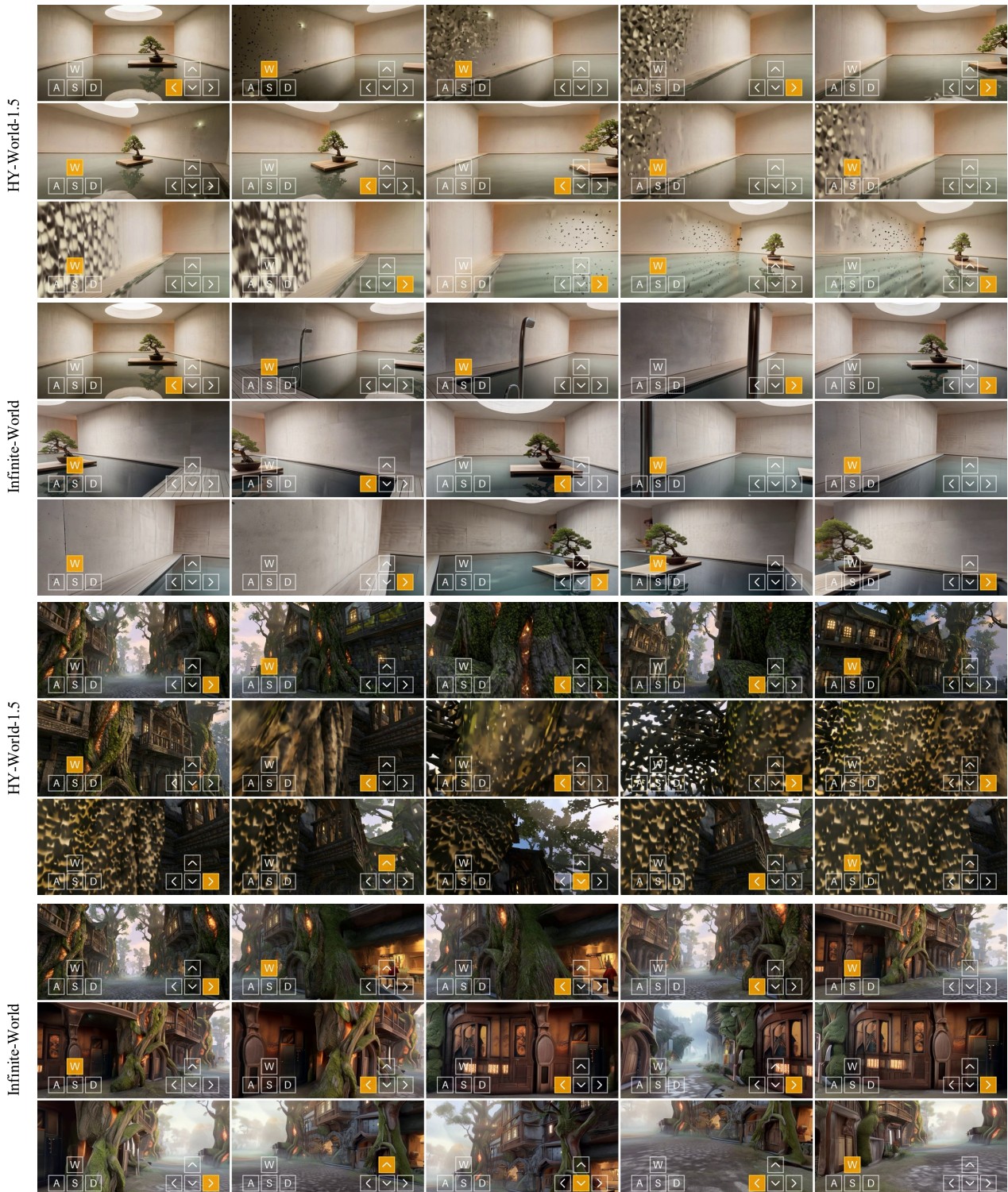

*Figure 7.* Visual comparison between Infinite-World and HY-World-1.5 on long-horizon range. **Zoom in for the best view.**

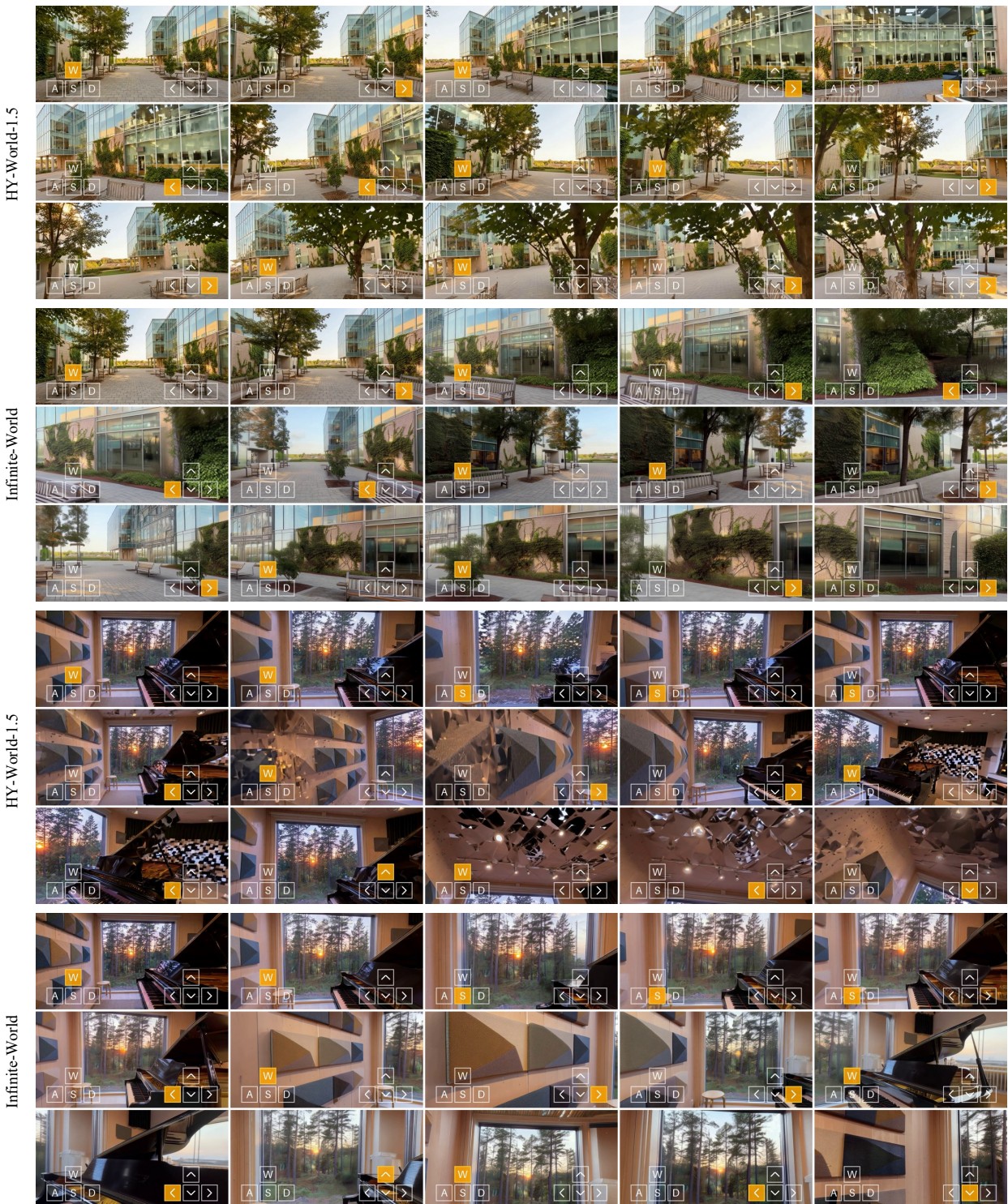

*Figure 8.* Visual comparison between Infinite-World and HY-World-1.5 on long-horizon range. **Zoom in for the best view.**

