# OpenReview forum: "Infinite-World: Scaling Interactive World Models to 1000-Frame Horizons via Pose-Free Hierarchical Memory"
_ICML.cc/2026/Conference — ICML 2026 regular_

### Official Review · Reviewer_25gJ · 2026-03-12

**Soundness:** 3
**Presentation:** 3
**Significance:** 2
**Originality:** 3
**Overall Recommendation:** 4
**Confidence:** 5

**Summary:**

This paper proposes Infinite-World, an interactive world model that achieves coherent 1000+ frame simulation in real-world environments, solving the reality gap issue of traditional models. Its core innovations are three key components: a pose-free hierarchical memory compressor with constant computational complexity, an uncertainty-aware tri-state action labeling module for noisy pose data, and a revisit-dense finetuning strategy that activates long-range loop-closure with only 30 minutes of data, based on pilot study insights about memory and topological density.

**Compliance With Llm Reviewing Policy:**

Affirmed.

**Final Justification:**

Thank the authors for the detailed rebuttal. Most of my concerns have been addressed. I will raise my rating to borderline acceptance.

**Key Questions For Authors:**

see the weakness.

**Limitations:**

While the authors do address certain technical limitations of the model in the conclusion, such as cumulative visual drift for horizons exceeding 1000 frames and the model’s reliance on the specific pre-trained Wan-2.1-1.3B backbone, their discussion is overly cursory and lacks in depth analysis into the underlying causes of these limitations.

**Strengths And Weaknesses:**

### Strengths:
1. HPMC innovatively designs a two-stage hierarchical recursive compression mechanism with a fixed budget, addressing the long-sequence computational bottleneck in a pose-free manner; UAL proposes a tri-state action labeling logic that retains uncertain samples, effectively adapting to pose estimation noise in real-world scenarios; pilot experiments explore the laws of long-range memory activation, and the revisit-dense finetuning strategy designed accordingly enables loop-closure capability activation with a small amount of data.

2. The discussion of computational efficiency stands out as a key merit of this paper. The core claim that hierarchical compression can effectively constrain memory footprint is well supported by Figure 5, which shows that the memory usage of the hierarchical compression scheme plateaus over time, whereas that of the non-compression and direct compression schemes increases continuously.

3. The paper features highly intuitive visualizations. The qualitative comparisons in Figure 4 and the long-horizon experimental examples in Figures 7 and 8 effectively illustrate the failure modes the authors aim to address, particularly visual drift and ghosting artifacts.


### Weaknesses:
1. Section 4.2.1’s benchmark uses generated initial images and handcrafted trajectories, with Table 1 quantitative results relying mostly on VBench quality metrics and subjective rankings. Long-horizon consistency is the key contribution, yet no objective metrics directly measure revisit accuracy, loop-closure fidelity, landmark preservation or geometric consistency over long returns; the only strong evidence is visual, which is prone to cherry-picking. The authors could adopt a reverse trajectory strategy to evaluate memory capability by using reconstruction metrics such as PSNR, SSIM and LPIPS.

2. the paper needs stronger evidence to justify its broader framing of breaking the barriers to 1000+ frame real-world simulation. At the moment, that case is not fully made.

3. The implementation sets N=5, a window size W=64, and a compression target of T_max=20, a configuration stated to support a context of up to 320 frames. Yet the paper’s title and abstract center on enabling 1000+ frame simulation, and nowhere in the manuscript is a precise algorithm provided for how recursion is scheduled in autoregressive generation beyond the 320-frame range. Critical details are also missing, including the number of compression levels employed, whether earlier compressed summaries undergo online re-compression, and the methods used to control error accumulation from repeated compression operations.

---

> ### Author Rebuttal · Authors · 2026-03-31
>
> We thank the reviewer for recognizing our HPMC design, the tri-state UAL action labeling, the RDD strategy, and the visualization of computational efficiency. The main concerns focus on three points: the lack of more direct objective evaluation for long-term memory, whether the “1000+ frame” claim is sufficiently supported, and whether the recursive scheduling mechanism of HPMC beyond a 320-frame context is clearly explained. We respond below.
>
> ### 1. Long-Term Memory Evaluation and the Evidence Boundary of the “1000+ Frame” Claim
>
> We fully agree that long-horizon consistency, as the core contribution of this paper, should not rely only on visual examples and subjective preference results. In the current version, we mainly use user studies, VBench, and long-horizon visualizations to show long-term memory performance, but these are not sufficient to directly quantify revisit accuracy or loop-closure fidelity. Following the reviewer’s suggestion, we have added more direct quantitative evaluation.
>
> We find that pixel-level metrics such as PSNR, SSIM, and LPIPS are not ideal for this purpose: even when two frames are highly consistent in 3D from a visual perspective, small viewpoint differences can cause a large drop in these scores, making them less reliable for measuring actual memory ability. We therefore use Met3R to evaluate 3D consistency. This metric measures cross-view 3D consistency between two images of the same scene from different viewpoints, where lower is better. Specifically, we compute Met3R between the first frame and the last frame of videos with different lengths. These results also support our claim that our method performs best at the 1000+ frame scale.
>
> | Method | 16 chunks | 8 chunks | 4 chunks |
> | --- | --- | --- | --- |
> | Hunyuan-GameCraft | 0.488 | 0.470 | 0.346 |
> | Matrix-Game 2.0 | 0.610 | 0.485 | 0.384 |
> | Yume 1.5 | 0.504 | 0.299 | 0.296 |
> | HY-World-1.5 | 0.491 | 0.272 | 0.173 |
> | Ours | 0.422 | 0.298 | 0.285 |
>
> Lower Met3R indicates better cross-view 3D consistency. At 16 chunks, our method performs best. HY-World-1.5 is stronger at short horizons partly because it uses a larger 8B backbone with better native visual quality, but its consistency drops much faster as the video becomes longer.
>
> ### 2. How a 320-Frame Context Supports 1000+ Frame Generation
>
> This is a very important point, and we will clarify it more explicitly. In the paper, the configuration `N=5`, `W=64`, and `T_max=20` corresponds to a history range of up to 320 latent frames in a single compression pass. Since the VAE used in Wan-2.1-1.3B has a 4x temporal compression ratio, this corresponds to 1280 video frames, which is why the setup can support 1000+ frame generation.
>
> More specifically, our recursive scheduling does not keep the full raw history throughout inference. Instead, it maintains a fixed-budget compressed memory. For relatively short histories, we directly apply the temporal encoder once. When the history exceeds the range that can be processed in a single pass, we first divide it into `N=5` overlapping windows, apply local compression to each window, concatenate these local summaries, and then apply a global compression step to obtain a fixed-memory representation with `T_max=20`. During subsequent autoregressive generation, this globally compressed memory is retained as the historical summary and is compressed again together with newly arrived history segments. Therefore, earlier compressed summaries are indeed recompressed online rather than stored forever in raw form. In terms of compression hierarchy, each update cycle consists of two levels, “local compression + global compression,” and this two-level structure is recursively applied as the rollout continues.
>
> ### 3. Error Accumulation and Limitations
>
> To mitigate the error accumulation caused by repeated compression, we mainly rely on three design choices. First, we use overlapping windows rather than hard partitions to reduce boundary information loss. Second, we explicitly preserve the most recent frame as local memory, so the model retains a reliable short-range visual anchor in addition to the long-term summary. Third, the compressor is jointly optimized with the generative backbone, so it learns to preserve the historical information that is most useful for future generation, rather than a generic reconstruction-oriented representation. That said, we also agree that error accumulation cannot be completely eliminated, and we still occasionally observe drift and scene forgetting at extremely long horizons. We will make the recursive scheduling pipeline clearer in the rebuttal and revision, and include concise pseudocode or a flow diagram to describe the implementation.

---

> > ### Author Rebuttal · Reviewer_25gJ · 2026-04-04
> >
> > Thank the authors for the detailed rebuttal. Most of my concerns have been addressed. I will raise my rating to borderline acceptance.

---

### Official Review · Reviewer_ZUJ6 · 2026-03-13

**Soundness:** 4
**Presentation:** 4
**Significance:** 4
**Originality:** 4
**Overall Recommendation:** 5
**Confidence:** 3

**Summary:**

This paper presents Infinite-World, an interactive world model designed to maintain coherent visual memory over simulations longer than 1000 frames in real-world environments. The paper focuses on two major challenges in real-world video modeling: noisy camera pose estimation and sparse viewpoint revisits. To address these issues, the authors propose three key components. First, a Hierarchical Pose-free Memory Compressor (HPMC) recursively compresses historical latents into a fixed memory budget, allowing long-horizon generation without explicit camera poses. Second, an Uncertainty-aware Action Labeling module discretizes continuous camera motion into a tri-state representation to filter noisy trajectories and improve action-response learning. Third, a Revisit-Dense Finetuning Strategy uses a curated 30-minute dataset to strengthen spatial memory and loop-closure behavior. Experiments show that the method can generate long interactive sequences with improved spatial consistency in real-world scenes.

**Compliance With Llm Reviewing Policy:**

Affirmed.

**Final Justification:**

I appreciate the authors' thorough explanations, which have successfully resolved my concerns. Therefore, I will keep the positive rating of this manuscript.

**Key Questions For Authors:**

Could the authors provide automated quantitative metrics, such as PSNR, SSIM, or LPIPS, to measure how closely generated loop-closure frames match the corresponding earlier reference frames?

How sensitive is the uncertainty-aware action labeling module to the choices of the noise threshold and the action trigger threshold?

The current HPMC uses a fixed memory budget of T_max=20. How does changing this budget affect the trade-off between long-horizon consistency and computational cost?

**Limitations:**

The paper does a reasonable job discussing both technical limitations and broader impacts. In particular, the authors acknowledge that the model can still accumulate drift and degrade visually over very long horizons, and they outline possible future directions such as self-forcing and improved noise schedules. The impact statement also appropriately notes the potential misuse risks of high-fidelity video generation systems.

**Strengths And Weaknesses:**

Strengths

The proposed HPMC is a well-motivated component that effectively addresses the memory growth problem in long-horizon generation. The reported results suggest that it keeps memory usage stable even for sequences up to 1300 frames, which is an important practical contribution.

The uncertainty-aware tri-state action labeling is also a sensible and pragmatic design choice. Given the unreliability of camera extrinsics estimated from raw videos, this mechanism provides a robust way to reduce noise in action supervision.

The paper is clearly organized, and the pilot study provides useful motivation for the overall data construction and training strategy. The visual comparisons in both the main paper and supplementary material are also convincing, showing stronger preservation of spatial landmarks over long horizons than competing methods.

The work is significant in that it pushes long-horizon interactive generation in real-world environments beyond 1000 frames. This is a meaningful step forward for world modeling, spatial intelligence, and embodied AI.

Weaknesses

A main weakness is that the evaluation of long-term memory consistency relies heavily on subjective human preference studies. The paper does not provide objective reconstruction-style metrics, such as PSNR, SSIM, or LPIPS, to directly measure how well loop-closure frames match previously seen views.

In addition, the architectural details of HPMC are somewhat limited. While the paper states that a 3D-ResNet is used to achieve temporal compression, the exact structure and layer design of this module are not described in enough detail for full clarity or reproducibility.

---

> ### Author Rebuttal · Authors · 2026-03-31
>
> ## Reviewer ZUJ6
>
> We thank the reviewer for the positive assessment of our idea and experiments, and for recognizing the practical value of HPMC, UAL, and RDD. The main questions concern more direct quantitative evaluation and clearer details and sensitivity analysis for HPMC and UAL. We respond below.
>
> ### 1. On Objective Quantitative Evaluation
>
> We find that pixel-level metrics such as PSNR, SSIM, and LPIPS are not ideal for evaluating long-term memory: even when the generated results appear highly consistent in 3D, small viewpoint differences can cause a large drop in these scores, making them less reliable for measuring actual memory ability.
>
> We therefore evaluate 3D consistency using Met3R, which measures cross-view consistency between two images of the same scene from different viewpoints, where lower is better. Specifically, we compute Met3R between the first frame and the last frame of videos with different lengths. In addition, we include 100 real-world images from DL3DV to further validate the model’s ability in real scenes.
>
> | Method | 16 chunks | 8 chunks | 4 chunks |
> | --- | --- | --- | --- |
> | Hunyuan-GameCraft | 0.488 | 0.470 | 0.346 |
> | Matrix-Game 2.0 | 0.610 | 0.485 | 0.384 |
> | Yume 1.5 | 0.504 | 0.299 | 0.296 |
> | HY-World-1.5 | 0.491 | 0.272 | 0.173 |
> | Ours | 0.422 | 0.298 | 0.285 |
>
> Lower Met3R indicates better cross-view 3D consistency. At 16 chunks, our method performs best. HY-World-1.5 is stronger at short horizons partly because it uses a larger 8B backbone with better native visual quality, but its consistency drops much faster as the video becomes longer.
>
> ### 2. On UAL Threshold Sensitivity
>
> For UAL, we use two thresholds, `\tau_1` (noise floor) and `\tau_2` (action trigger), to divide motion magnitude into three states: `No-operation / Discrete Action / Uncertain`. Here, `Uncertain` is independently encoded as an action label, similar to actions such as `go forward` or `turn left`. As for threshold selection, we did not tune them by repeatedly rerunning large-scale pretraining, since that would be prohibitively expensive. Instead, we determined suitable thresholds through a small-scale user study that measures action-label accuracy under different threshold settings.
>
> | $\tau^{trans}$ | $\tau^{rot}$ | Accuracy |
> | --- | --- | --- |
> | 0.008 | 1.5 | 80.6% |
> | (0.005, 0.01) | (1.25, 1.75) | 87.1% |
> | (0.002, 0.015) | (1.0, 2.0) | 91.9% |
>
> Expanding the range of the `Uncertain` state may further improve label accuracy, but it would also reduce the amount of data assigned to meaningful action labels. In our experiments, the parameter setting in the last row achieves a desirable trade-off.
>
> ### 3. On the Trade-off Between `T_max` and Long-Horizon Consistency
>
> Your question about the fixed memory budget `T_max=20` is very important. For our target horizon (16 chunks, approximately 320 latent frames), `T_max=20` is the largest compression ratio achieved after two stages of compression. We additionally test `T_max=25` and `T_max=30`, and evaluate both long-video performance using Met3R.
>
> | `T_max` | Met3R $\downarrow$|
> | --- | --- |
> | 20 | 0.422 |
> | 25 | 0.408 |
> | 30 | 0.428 |
>
> These results show that `T_max=25` achieves the best performance. Although increasing `T_max` beyond 25 introduces a larger memory budget, the additional redundancy may interfere with effective summarization, which likely explains why `T_max=30` performs worse. Compared with `T_max=20`, `T_max=25` provides better accuracy, though at a slower processing speed. In our future open-source release, we will provide recommended parameter settings for different user needs and resource constraints.
>
> ### 4. On HPMC Architecture Details
>
> We agree that the current manuscript does not describe HPMC in sufficient detail, especially the 3D-ResNet compressor. In our implementation, the temporal encoder is a 3D-ResNet-style module: the input latents are first projected by a `3×3×3` `Conv3d`, followed by two down blocks. Each down block contains two `ResnetBlock3D` layers and one `TemporalDownsample` module that downsamples only along time, preserving spatial resolution. After these two stages, the total temporal compression ratio is `4×`. In the bottleneck, we use a mid block of `ResBlock + Wan-style self-attention + ResBlock`, where the attention adopts the same 3D RoPE and FlashAttention design as the backbone to preserve important cross-time and cross-view cues. The features are then projected back to the original latent channel dimension by `GroupNorm + SiLU + Conv3d`. For long histories, HPMC first compresses local sliding-window segments and then applies a second global compression over the concatenated summaries, forming a hierarchical recursive structure of “local compression + global compression.” We will clarify these details in the revision and appendix, and release the full implementation code to improve clarity and reproducibility.

---

> > ### Author Rebuttal · Reviewer_ZUJ6 · 2026-04-02
> >
> > I appreciate the authors' thorough explanations, which have successfully resolved my concerns. Therefore, I lean towards recommending the acceptance of this manuscript.

---

### Official Review · Reviewer_EUHG · 2026-03-15

**Soundness:** 3
**Presentation:** 2
**Significance:** 2
**Originality:** 2
**Overall Recommendation:** 4
**Confidence:** 3

**Summary:**

This paper proposes Infinite-World, an interactive world model designed to support long-horizon simulation (1000+ frames) using real-world video data. The authors introduce a hierarchical pose-free memory compressor to maintain long-term context with bounded computation, an uncertainty-aware action labeling mechanism to handle noisy motion signals, and a revisit-dense finetuning strategy to activate spatial memory. Experiments on a long-horizon interactive benchmark suggest improved visual consistency, action responsiveness, and memory stability compared with several recent interactive world models.

**Compliance With Llm Reviewing Policy:**

Affirmed.

**Final Justification:**

The authors have addressed all my concerns.

**Key Questions For Authors:**

None

**Strengths And Weaknesses:**

### Strengths

1. The paper targets the challenge of long-horizon interactive world modeling, which is relevant for applications such as embodied AI, simulation, and generative environments.

2. The proposed approach integrates memory compression, action labeling, and a tailored training strategy into a coherent architecture for long-horizon interactive video generation.

3. The experiments include both automated metrics and human user studies to evaluate visual quality, memory consistency, and action responsiveness.

4. The results suggest that the proposed method can maintain more stable visual consistency over extended horizons compared with several recent baselines.


### Weaknesses

1. **Limited methodological novelty.**
   The proposed framework mainly integrates several existing ideas—hierarchical memory compression, motion discretization, and dataset-specific finetuning—into a unified system. Related techniques such as long-context compression, frame packing, and memory tokens have already been explored in the long-video generation and world model literature. As a result, the contribution appears more like a system-level engineering integration rather than a fundamentally new modeling paradigm or algorithmic breakthrough. The paper also lacks deeper theoretical or empirical analysis explaining why the particular combination of these components leads to substantial capability improvements.

2. **Mismatch between the central claim and the experimental scale.**
   The paper claims stable **1000+ frame interactive world simulation**, which is a strong claim. However, the experimental scale is relatively limited. The benchmark contains only **100 scenarios**, the action trajectories are manually designed rather than generated from real interactions, and the environment distribution is relatively controlled. Under such conditions, it is difficult to assess whether the model would maintain similar performance in more complex or realistic interaction environments.

3. **Potential bias in the benchmark design.**
   The benchmark is constructed using **text prompts, generated initial frames, and predefined action trajectories**. This setup introduces several limitations:
   - The initial environment is synthesized by an image generation model rather than captured from real scenes.
   - The action sequences are manually scripted instead of arising from real agent–environment interactions.
   - The setup lacks closed-loop interaction between the agent and the environment.
   Therefore, the evaluation may resemble a controlled video generation test rather than a realistic interactive world modeling scenario.

4. **Baseline comparisons are not fully controlled.**
   The baseline comparisons involve several confounding factors. Some baselines are trained on different data distributions (synthetic vs real-world), the parameter scales of the models differ significantly, and the training procedures or context lengths may not be consistent across methods. In addition, it is unclear whether all baselines receive comparable finetuning or SFT-style training. These factors make it difficult to attribute the reported improvements solely to the proposed method.

5. **Strong dependence on a specialized dataset strategy.**
   The RDD finetuning dataset plays a key role in achieving the reported performance gains. However, this dataset requires specially curated revisit-dense trajectories and controlled capture conditions. As a result, the improvements may largely stem from dataset design rather than the modeling approach itself, raising concerns about generalizability to broader real-world video data.

6. **Evaluation metrics do not directly verify world modeling capability.**
   The improvements on VBench metrics are relatively modest, and much of the claimed advantage comes from user study results, which are inherently subjective. Moreover, these metrics do not directly measure properties expected from a world model, such as geometric consistency, spatial structure preservation, or long-term environmental coherence. It therefore remains unclear whether the model truly learns a coherent world representation.

7. **Reproducibility and training cost concerns.**
   Several important implementation details are insufficiently specified, including the action label generation pipeline (e.g., the accuracy of VGGT-based pose estimation), dataset filtering and sampling strategies, and full training configurations. In addition, the method requires substantial GPU resources, but the paper does not clearly report training cost or resource requirements, which may limit reproducibility.

8. **Potentially overstated framing of “Infinite-World.”**
   The paper frames the system as “Infinite-World,” while the demonstrated capability corresponds to approximately **1000-frame simulations**. Although this is a notable horizon, the naming may appear somewhat exaggerated relative to the actual scale demonstrated in the experiments.

9. **Possible inconsistency in implementation details.**
   In Section 4.1.1 (Implementation Details), the backbone model is described as **Wanx-2.1-1.3B**, which appears unusual and may be a typographical error (possibly referring to Wan-2.1-1.3B). Such inconsistencies make it difficult to precisely understand the experimental setup.

---

> ### Author Rebuttal · Authors · 2026-03-31
>
> We thank for effort the reviewer EUHG. The main concerns are about methodological novelty, the realism of the experimental setup, fairness of comparison, and reproducibility. We respond to these points below.
>
> ### 1. Methodological Novelty (Weaknesses 1)
> We agree that this work is not intended to propose a completely new modeling paradigm detached from existing generative-model frameworks. Rather, the main contribution of our paper is a practical training and modeling pipeline for long-horizon interactive world modeling in real-world data.
>
> The novelty does not come from an isolated module alone, but from addressing several tightly coupled bottlenecks in real-world settings:
> - Unreliable noisy pose estimation
> - Sparse viewpoint revisits in natural videos;
> - Existing methods lack a pose-free memory mechanism for long-horizon generation.
>
> Our HPMC and UAL address these issues in a new way, and prior work does not provide the same type of memory mechanism or action-labeling strategy. In addition, the toy experiments used to motivate our RDD design provide a novel perspective on how long-range memory can be activated in real-world training.
>
> ### 2. Benchmark Limitations (Weaknesses 2, 3, 6)
> - Lack of real interaction: our model is an action-conditioned world model and does not include autonomous agent-environment interaction. The reviewer may have interpreted it as a world action model, which corresponds to a different line of work. Similar to previous works such as Hunyuan-GameCraft and HY-WorldPlay, we design diverse action trajectories so that we can simultaneously evaluate visual quality, memory consistency, and action responsiveness.
> - Initial images generated by a model: our benchmark is built from real video data in DL3DV. We additionally sampled 100 first frames from real videos as benchmark starts. These data cover diverse real-world scenes, and we will further expand the benchmark size in the revision.
> - Lack of direct 3D-consistency metrics: we have additionally evaluated the new benchmark using Met3R. This allows us to measure consistency across different context lengths for each model. Met3R is used here to quantify cross-view consistency within the same scene.
>
> | Method | 16 chunks | 8 chunks | 4 chunks |
> | --- | --- | --- | --- |
> | Hunyuan-GameCraft | 0.488 | 0.470 | 0.346 |
> | Matrix-Game 2.0 | 0.610 | 0.485 | 0.384 |
> | Yume 1.5 | 0.504 | 0.299 | 0.296 |
> | HY-World-1.5 | 0.491 | 0.272 | 0.173 |
> | Ours | 0.422 | 0.298 | 0.285 |
>
> Lower Met3R indicates better cross-view 3D consistency. Under the longest horizon of 16 chunks, our method achieves the best result among all compared methods. HY-World-1.5 achieves stronger short-horizon consistency because they use a larger 8B backbone, which provides better native visual quality. However, as the video horizon increases, its consistency degrades much more rapidly, while our method remains more stable over long rollouts.
>
> ### 3. Baseline Fairness (Weaknesses 4)
> - The compared baselines do not release full training code, so we use the best available settings and official checkpoints provided by the authors whenever possible.
> - Our training pipeline and data construction are part of the contribution of this work, so they should not be artificially separated from the final performance comparison.
>
> ### 4. Dependence on RDD (Weaknesses 5)
> We agree that RDD plays a key role in the final performance, but we do not believe this weakens the contribution of the work. Instead, it supports one of our central conclusions: for long-horizon spatial memory in real-world data, the decisive factor is the temporal duration and topological revisit density of trajectories but not data volume. Our ablations show that RDD serves to activate long-range loop-closure capability, while HPMC and UAL are responsible for maintaining long-term context under a fixed budget and enabling robust action-conditioned learning under noisy poses. Therefore, the performance gain does not simply come from “more data,” but from a data construction strategy and model design.
>
> ### 5. Reproducibility, Training Cost, and Implementation Details (Weaknesses 7)
> - VGGT annotation: we sample 5 frames per second and determine the action label from displacement and rotation within each second. We use 8 H100 GPUs, and labeling about 30 hours of video requires roughly 40 H100 GPU hours.
> - Data filtering: we simply filter out video intros and outros. We use Qwen-2.5-VL-32B for caption annotation, producing one caption every 5 seconds.
> - Training configuration: in addition to what is already reported, we will supplement the following details: AdamW optimizer, learning rate `1e-5`, weight decay `1e-2`, and a total training cost of about 3000 H100 GPU hours using 16 H100 GPUs.
> - We will release the code to improve reproducibility.
>
> ### 6. Writing Issues (Weaknesses 8, 9)
> We will revise the method naming to make the presentation more precise, and we will correct the backbone name to `Wan-2.1-1.3B`.

---

> > ### Author Rebuttal · Reviewer_EUHG · 2026-04-03
> >
> > The authors have addressed all my concerns.

---

### Official Review · Reviewer_GwpT · 2026-03-24

**Soundness:** 2
**Presentation:** 3
**Significance:** 3
**Originality:** 3
**Overall Recommendation:** 4
**Confidence:** 4

**Summary:**

This paper studies long-horizon interactive world modeling in real-world videos, where noisy pose estimation, scarce viewpoint revisits, and the high cost of long-context memory make controllable simulation difficult. The authors propose Infinite-World, which combines a Hierarchical Pose-free Memory Compressor for fixed-budget long-range context modeling, Uncertainty-aware Action Labeling for more robust action supervision under noisy trajectories, and a Revisit-Dense Finetuning strategy to activate long-range loop-closure behavior with a compact dataset. Experiments on a long-horizon benchmark and user studies suggest that the method improves memory consistency, visual fidelity, and action responsiveness compared with several recent baselines.

**Compliance With Llm Reviewing Policy:**

Affirmed.

**Final Justification:**

Most of my concerns have been resolved through rebuttal. I will have a positive rating on the paper.

**Key Questions For Authors:**

1. The method compresses increasingly long histories into a fixed memory budget $T_{\max}$. How does this design behave when the trajectory becomes much longer or when the scene changes more substantially over time? In particular, does forcing highly variable long-horizon content into a fixed memory length lead to noticeable loss of detail or memory fidelity? It would also be helpful to discuss, under a given $T_{\max}$, how far the model can be pushed in terms of inference horizon before the generation becomes unstable or memory consistency degrades significantly.

2. UAL relies on thresholds (e.g., $\tau_1$ and $\tau_2$) to distinguish no-op, discrete, and uncertain actions. How are these thresholds selected in practice, and how sensitive are the results to them? In particular, do they need to be retuned across environments with different motion statistics, camera dynamics, or action magnitudes?

3. The paper introduces an "uncertain" action category, but it is not fully clear how these samples are handled during training and inference. Are they filtered out, explicitly encoded as a separate action state, or treated differently in the loss/objective?

**Limitations:**

Partially. The paper briefly acknowledges technical limitations, but the discussion could be more complete. In particular, it would be helpful to more explicitly discuss how the fixed-memory design behaves under much longer trajectories or scenes with substantial changes over time, and how well the current evaluation setup, which is relatively controlled, is expected to reflect more natural and open-ended real-world interaction scenarios.

**Strengths And Weaknesses:**

**Strengths:**
- The proposed framework is well structured, with the three main components playing complementary roles: HPMC targets scalable long-range memory, UAL improves robustness of action supervision under noisy trajectories, and RDD finetuning is designed to activate long-range loop-closure behavior.
- The empirical section is reasonably comprehensive. In addition to quantitative metrics, the paper includes user studies, qualitative comparisons, and ablations, which together provide a useful picture of the method’s practical behavior.
- The idea of using a compact revisit-dense finetuning set to strengthen long-range spatial memory is interesting and potentially impactful for real-world interactive world modeling, where revisit structure is often more important than raw data volume.

**Weaknesses:**
- The robustness of several key design choices is not sufficiently analyzed. In particular, the paper does not provide a sensitivity study for the HPMC memory-partition hyperparameters (e.g., number of chunks, window size, and stride behavior), even though these parameters directly affect the effective memory/compression trade-off. Similarly, UAL appears to depend on threshold choices (e.g., $\tau_1$ and $\tau_2$) for separating no-op, discrete, and uncertain actions, but the paper does not clearly discuss how these thresholds are selected, how sensitive performance is to them, or whether they transfer across environments with different motion statistics.
- The treatment of the "uncertain" action state is not fully clear. It is difficult to tell whether such cases are filtered, explicitly encoded, or supervised in a different way during training and inference. Since this is a core part of the proposed action-labeling scheme, a clearer explanation would strengthen both reproducibility and technical clarity.
- Although HPMC is shown to improve memory efficiency, the paper does not thoroughly characterize the quality–compression trade-off under a fixed $T_{\max}$, especially as the number of past chunks grows or as scene variation becomes larger. Relatedly, it would be useful to better understand how far the fixed-memory design can be pushed before memory fidelity degrades substantially.
- The paper is generally readable, but a few presentation details could be improved. For example, some paragraph-level subheadings in Section 3.3 visually resemble higher-level section headings, which slightly interrupts the flow of reading.

---

> ### Author Rebuttal · Authors · 2026-03-31
>
> We thank the reviewer for recognizing the overall structure of our framework, the completeness of the experiments, and the value of the RDD design. The questions raised are highly important, and we clarify them below.
>
> ### 1. HPMC Sensitivity and Fixed Memory Budget
>
> Regarding the HPMC hyperparameters, we partition the historical latents into `N=5` overlapping windows (`W=64`, with dynamic stride `S` determined by the total history length), perform local compression first, and then apply global compression, finally compressing the history into a fixed memory budget of `T_max=20`. This configuration is chosen to support our target of 1000+ frame memory consistency: the maximum temporal coverage of the windows is `64 x 5 = 320` latent frames, corresponding to 1280 video frames. To study the effect of `W` and `N`, we additionally varied these settings and measured 3D consistency under different configurations using Met3R.
>
> Met3R quantifies cross-view consistency within the same scene across different context lengths. The lower denotes better.
>
> | Window Size `W` | Number of Windows `N` | Max Covered Latent Frames | Met3R $\downarrow$ |
> | --- | --- | --- | --- |
> | 64 | 5 | 320 | 0.422 |
> | 80 | 4 | 320 | 0.428 |
> | 40 | 8 | 320 | 0.453 |
>
> As illustrated in the table, `W=64m N=5` is the best parameter group, which is also our final choice.
>
> We further conduct the ablation study on `T_max`. For our target horizon (16 chunks, approximately 320 latent frames), `T_max=20` is the largest compression ratio achieved after two stages of compression. We additionally test `T_max=25` and `T_max=30`, and evaluate both long-video performance using Met3R.
>
> | `T_max` | Met3R $\downarrow$|
> | --- | --- |
> | 20 | 0.422 |
> | 25 | 0.408 |
> | 30 | 0.428 |
>
> These results show that `T_max=25` achieves the best performance. Although increasing `T_max` beyond 25 introduces a larger memory budget, the additional redundancy may interfere with effective summarization, which likely explains why `T_max=30` performs worse. Compared with `T_max=20`, `T_max=25` provides better accuracy, though at a slower processing speed. In our future open-source release, we will provide recommended parameter settings for different user needs and resource constraints.
>
> ### 2. UAL Thresholds and the `Uncertain` State
>
> For UAL, we use two thresholds, $\tau_1$ (noise floor) and $\tau_2$ (action trigger), to divide motion magnitude into three states: `No-operation / Discrete Action / Uncertain`. Here, `Uncertain` is independently encoded as an action label, similar to actions such as `go forward` or `turn left`. As for threshold selection, we did not tune them by repeatedly rerunning large-scale pretraining, since that would be prohibitively expensive. Instead, we determined suitable thresholds through a small-scale user study that evaluates action-label accuracy under different threshold settings.
>
> | $\tau^{trans}$ | $\tau^{rot}$ | Accuracy |
> | --- | --- | --- |
> | 0.008 | 1.5 | 80.6% |
> | (0.005, 0.01) | (1.25, 1.75) | 87.1% |
> | (0.002, 0.015) | (1.0, 2.0) | 91.9% |
>
> Expanding the range of the `Uncertain` state may further improve label accuracy, but it would also reduce the amount of data assigned to meaningful action labels. In our experiments, the parameter setting in the last row achieves a desirable performance.
>
> ### 3. Boundary of Inference Horizon Under a Fixed Budget
>
> Regarding the boundary of the inference horizon, our paper takes a relatively cautious position. Our pilot study already shows that long-horizon stability is tightly coupled with the temporal window seen during training, and obvious drift or collapse may occur once inference goes substantially beyond the training context. Therefore, the contribution of Infinite-World is to significantly push this boundary forward through HPMC and RDD, enabling coherent generation over 1000+ frames under the current setting. Beyond this range, we do observe visually noticeable scene forgetting.

---

> > ### Author Rebuttal · Reviewer_GwpT · 2026-04-03
> >
> > Most of my concerns have been resolved. I will keep my rating positive.

---

### Decision · Program_Chairs · 2026-04-30

**Decision:**

Accept (regular)

**Comment:**

This paper received all positive reviews (3 WA, 1 A) with the reviewers generally praising the paper's problem area, overall solution (the three elements of pose-aware memory compression, action-labeling, and revisit dense finetuning) and the experimental results (both the qualitative and quantitive). The main weaknesses listed are the precise technical novelty of the paper (in that most of the technical elements are taken from prior work) and the thoroughness of the evaluation (reviewers noted that the paper should go beyond just user studies and general quality metrics). I tend to agree with reviewers on both the strengths and weaknesses of the paper and advocate for acceptance. I would encourage the authors to integrate the additional evaluations presented during the rebuttal into the paper.